# Towards remote monitoring in pediatric care and clinical trials—Tolerability, repeatability and reference values of candidate digital endpoints derived from physical activity, heart rate and sleep in healthy children

**M. D. Kruizinga**[1,2,3]*, **N. van der Heide**[1,2], **A. Moll**[1,2], **A. Zhuparris**[1], **Y. Yavuz**[1], **M. L. de Kam**[1], **F. E. Stuurman**[1,3], **A. F. Cohen**[1,3], **G. J. A. Driessen**[2,4]

**1** Centre for Human Drug Research, Leiden, The Netherlands, **2** Juliana Children's Hospital, HAGA Teaching Hospital, The Hague, The Netherlands, **3** Leiden University Medical Centre, Leiden, The Netherlands, **4** Maastricht University Medical Centre, Maastricht, The Netherlands

* mkruizinga@chdr.nl

## Abstract

### Background

Digital devices and wearables allow for the measurement of a wide range of health-related parameters in a non-invasive manner, which may be particularly valuable in pediatrics. Incorporation of such parameters in clinical trials or care as digital endpoint could reduce the burden for children and their parents but requires clinical validation in the target population. This study aims to determine the tolerability, repeatability, and reference values of novel digital endpoints in healthy children.

### Methods

Apparently healthy children (n = 175, 46% male) aged 2–16 were included. Subjects were monitored for 21 days using a home-monitoring platform with several devices (smartwatch, spirometer, thermometer, blood pressure monitor, scales). Endpoints were analyzed with a mixed effects model, assessing variables that explained within- and between-subject variability. Endpoints based on physical activity, heart rate, and sleep-related parameters were included in the analysis. For physical-activity-related endpoints, a sample size needed to detect a 15% increase was calculated.

### Findings

Median compliance was 94%. Variability in each physical activity-related candidate endpoint was explained by age, sex, watch wear time, rain duration per day, average ambient temperature, and population density of the city of residence. Estimated sample sizes for candidate endpoints ranged from 33–110 per group. Daytime heart rate, nocturnal heart rate and

**Data Availability Statement:** All relevant data are within the paper and its Supporting information files.

**Funding:** The author(s) received no specific funding for this work.

**Competing interests:** The authors have declared that no competing interests exist.

**Abbreviations:** AIC, Akaike Information Criterion; BMI, Body Mass Index; BP, Blood pressure; CV, Coefficient of variability; HR, Heart rate; ICC, intraclass correlation coefficient; IQR, Interquartile range; MVPA, Moderate-to-vigorous physical activity; PA, Physical activity; PPG, Photo plethysmography; QoL, Quality of life; SD, Standard deviation.

sleep duration decreased as a function of age and were comparable to reference values published in the literature.

## Conclusions

Wearable- and portable devices are tolerable for pediatric subjects. The raw data, models and reference values presented here can be used to guide further validation and, in the future, clinical trial designs involving the included measures.

## Introduction

Despite several initiatives to stimulate pediatric clinical trial initiation and conduct, the proportion of pediatric trials is 9% [1], even though children are the recipient of 25% of the global disease burden [2]. Recruitment and ethical barriers are often cited as a cause by investigators, while fear for invasive measurements and logistical difficulties, such as school schedules for children and demanding jobs for parents are burdens on the participant's side [3]. However, the improvement of wireless and portable technology may allow decentralized trials by using health devices in a home-setting. The digital endpoints included in such trials could significantly reduce the burden for children and their parents.

Before integration in clinical trials, digital endpoints must be validated fit-for-purpose [4]. During clinical validation, the tolerability and repeatability of the candidate endpoint must be determined. Furthermore, an important criterion for novel endpoints is a clinically significant difference between patients and healthy controls. In order to determine this in pediatric subjects, data from a large healthy cohort with a wide age range is paramount. While multiple feasibility studies have been conducted with wearables in various age-groups [5–7], no large-scale studies with a wide age-range have been conducted to investigate the measurements in a home-setting. Additionally, there is little pediatric data available regarding the impact of variability that is introduced by performing measurements in free-living conditions. Factors such as school, weather, city of residence and regularly playing sports are likely to influence digitally captured measures like physical activity and heart rate (HR).

It is established that chronically ill children exhibit a lower activity level compared to their healthy peers, and it is plausible that easy to obtain measurements like physical activity, HR, or sleep parameters are correlated to symptom severity in both acutely- and chronically ill children [8]. However, there is a need to translate raw wearable- and sensor data into novel clearly defined endpoints that are validated and sensitive to detect a change in disease severity in children [9]. Validated non-invasive digital endpoints with proven worth for the purpose of monitoring disease-activity could then not only be used in clinical research, but in clinical care as well. For example, home-monitoring of asthma symptoms via an electronic questionnaire has been shown to reduce the need for outpatient clinic visits [10], and including objective measurements with wearable- or portable devices may improve the reliability of home-monitoring even more.

The aim of this study is to investigate the tolerability of a combination of digital health devices via a remote clinical trial regimen, to obtain reference values in healthy children for said devices, to assess conditions that induce variability in free-living conditions and to explore novel features that could be used as candidate digital endpoint in future clinical trials.

## Materials & methods

### Location and ethics

This study was conducted at the Juliana Children's Hospital (HAGA teaching hospital, the Hague, the Netherlands) and the Centre for Human Drug Research (Leiden, the Netherlands) from November 2018 until March 2020. The study protocol was approved by the Medical Ethics Committee ZuidWest Holland (the Hague, the Netherlands) prior to initiation of the study. The study was conducted in compliance with the Dutch Act on Medical Research Involving Human Subjects (WMO) and Good Clinical Practice. Written informed consent was obtained from all parents and from children aged 12 years and older. Assent was obtained from children aged younger than 12. The trial was registered at the Dutch Trial Registry (NTR, Trial NL7612, registered 18-Mar-2019).

### Subjects and study design

Between 10–15 healthy children of each age year between 2 and 16 years old were recruited from the region around the Hague using various recruitment methods (local newspaper advertisement, distributing flyers in the Juliana Children's Hospital and on a local primary school). Exclusion criteria were the presence of (chronic) disease, inability to communicate in Dutch or English and an inability to use the devices. All children were visited at home by a trained investigator for a 30-minute training session and a baseline questionnaire. Children used the devices and completed a daily activity questionnaire for the subsequent 21 days, after which an end-of-study questionnaire was completed, and the devices were retrieved by the investigators. Children received a gift certificate worth €20 for their participation.

### Devices

Subjects wore a Steel HR smartwatch (Withings, Issy-les-Molineux, France) during the study period. The watch measures physical activity with a built-in accelerometer. HR was measured every 10 minutes via a photo plethysmography (PPG) sensor on the back of the watch. Furthermore, the watch calculates several sleep-related parameters using the accelerometer and an incorporated temperature sensor. All subjects performed twice weekly temperature measurements with the Withings Thermo device. Subjects ≥ 6 years old performed daily blood pressure (BP) measurements with the Withings BPM, a weekly weight measurement with the Withings Body+ scales and twice-weekly home-based spirometry using the Air Next spirometry device (NuvoAir, Stockholm, Sweden). The Air Next device employs a turbine mechanism with disposable mouthpieces and has been validated for use in children [11]. All devices used Bluetooth to connect to a smartphone (Motorola G6 (Motorola, Chicago, IL, USA)). The smartphone had the Withings Healthmate application, the CHDR MORE® application (used for data collection and aggregation) and an electronic patient reported outcome (ePRO) application pre-installed.

**Baseline- and environmental data.** Parents of all subjects completed the PedsQL 4.0 questionnaire and provided several baseline variables at the start of the study [12]. The population density of the Children's city of residence was classified using publicly available data of the Dutch Central Office of Statistics. Local weather statistics from a local weather station (Hoek van Holland, the Netherlands) were obtained from the Royal Dutch Meteorological Institute.

### Analysis

**Baseline characteristics and compliance.** Baseline characteristics were summarized. Compliance, an important indicator of the tolerability of the trial regimen, was calculated for each subject individually by dividing the amount of observations in the dataset by the amount

of expected observations (calculated per assessment). The same calculation was performed for all assessments together to calculate an overall compliance per subject. For the measurements that were not performed daily, a subject was considered noncompliant when a measurement time point deviated more than 1 day for spirometry and temperature assessments and more than two days for weight assessments. The median and interquartile range of the compliance within the complete cohort was calculated for each assessment and for the overall compliance. For assessments performed daily, the compliance over time was estimated by calculating the compliance for each individual study day (day 1 through day 21). The proportion of daily watch wear time per day was calculated using aggregated data per hour. An hour with no registered HR and step count data counted as noncompliant ('not worn'), and an hour with either a registered HR or a registered step counted as compliant ('worn'). The proportion of the wear time between 6AM and 10PM was calculated to include as a covariate when analyzing step count data. Data from screening days (n = 175) and all days with a watch wear time < 50% between 6AM and 10PM were excluded from analyses regarding daytime measurements (146 study days), while all days with a watch wear time < 50% between 0AM and 5AM (268 study days) were excluded from analyses regarding nocturnal measurements. This 50% threshold has been chosen in earlier studies as well [13, 14].

**Statistical analysis of candidate endpoints.** Candidate endpoints were analyzed with a linear mixed effects model with subject as random factor. Factors that were expected to explain variability were considered as fixed factor or covariate. Spline regression with 2 or 3 degrees of freedom was considered when nonlinear relationships were a possibility. Contribution to model fit was assessed by comparing the Akaike information criterion (AIC) of models and by likelihood ratio tests. A p-value smaller than 0.05 was considered statistically significant. Due to the exploratory nature of the study, no adjustment for multiple comparison was performed. Instead, covariate inclusion was guided by appraising the ΔAIC between candidate models. Interactions between factors or covariates were considered when biologically plausible. Residual plots were inspected for heteroscedasticity and non-normality and logarithmic or square root transformations were considered when assumptions were violated. However, mild non-normality of residuals was accepted due to the large size of the dataset [15]. Marginal effects and the 95% confidence interval of the effect were plotted for each variable in the final model. For selected candidate endpoints, a 90% prediction interval was constructed including random effect variance, in order to display cut-off values for the lowest 5% of measurements. The repeatability of each candidate endpoint was assessed by estimating the intra-subject coefficient of variability (CV). The CV was estimated by taking the square root of the residual variability of each model and dividing it by the estimated population mean.

**Physical activity.** Several candidate endpoints related to physical activity (measured by step count) were defined prior to analysis. These were divided in measurements per day and measurements per week. For daily measurements, step count per day (Daily PA) and step count during the most active hour per day (Daily $PA^{max}$) were defined. For weekly measurements, average step count per week, $10^{th}$ and $90^{th}$ centile of step count per day, and the $50^{th}$ and $90^{th}$ centile of step count per hour (between 6AM-10PM) was chosen. Parameters based on high and low centiles were chosen because peak-, trough- and average activity may exhibit different relationships with disease activity in patients. Factors that were hypothesized to explain variability were included as potential fixed factors or covariates during the analyses. The following parameters were considered: age, sex, body mass index (BMI), quality of life (QoL), rain duration, average temperature, wind speed, proportion of wear time between 6AM-10PM, population density of the subjects' city of residence, day of the week and school-day. A sample size needed to detect a 15% increase with 80% power was calculated for each candidate endpoint. Here, it was assumed that a 15% increase was clinically relevant, and that

the calculation was for a hypothetical study with parallel group design and follow-up period of 21 days.

**Heart rate.**    Average HR was summarized as average daytime HR (6AM– 10PM) and average nighttime HR (0AM-5AM). Age and sex were considered as covariate and factor, respectively. Estimated mean daytime HR was compared to the 10th-90th centile of reference values obtained from a recent meta-analysis regarding pediatric HR [16]. The relationship between daytime HR and physical activity was explored by including an interaction between step count and age in a separate model.

**Accelerometer-derived sleep parameters.**    Total sleep duration, sleep depth and amount of times a subject wakes up (wakeup count) were calculated by the Withings application and the Steel HR. All three were considered as candidate endpoint. Days with a sleep duration shorter than 3 hours and longer than 16 hours were excluded from the analysis as likely inaccurate measurement in this healthy cohort, considering the limitations of the Steel HR watch and published reference values in pediatrics [17]. Wakeup count was analyzed assuming a negative binomial distribution. Age, sex, school day and average ambient temperature were considered as independent variables in the models.

**Spirometry, blood pressure and temperature.**    All spirometry sessions were graded according to ATS/ERS criteria [18]. Spirometry sessions graded D or worse were excluded from further analyses. FEV1 and FVC (% of predicted) were summarized by age. Temperature and BP measurements were graphically displayed.

**Software and data pipeline.**    Data collected via the Withings devices was automatically transferred to the Withings servers, based on protocols maintained by Withings. A validated data pipeline requested the data from the Withings server and stored it on a Microsoft Azure Datalake (Microsoft, Redmond, WA, USA). From here, PySpark version 2.4.6 was used for data aggregation and tabulation. R version 3.5.1 and the lme4, emmeans, ggeffects and pwr packages were used for statistical analysis.

## Results

### Baseline characteristics

175 children were included. Baseline characteristics are displayed in Table 1. 45.7% of children was male. 85% of children practiced some type of sports. The average QoL score measured by the PedsQL questionnaire was 90.7 out of 100 (IQR [86–97]).

### Compliance and tolerability

The average compliance of each individual measurement is listed in Table 2. Median overall compliance was 94% (IQR 87–97%) of expected measurements. Median watch wear time was 23.6 hours per day. Lowest compliance was seen for spirometry and temperature measurements (median 83%). Subjects aged 2 or 3 years exhibited a slightly lower overall compliance (S1 Fig), and 11 subjects (6%) exhibited an overall compliance lower than 70%, including two children (aged 3) who stopped participation due to being uncomfortable wearing the watch continuously. There was no correlation between age and compliance for any of the measurements. Compliance appeared to decrease over time for blood pressure- and questionnaire assessments (S2 Fig).

160 (91%) of all subjects completed the EOS questionnaire. Of these, subjects reported to have spent 8.5 (SD 5.5) minutes per day on study-related assessments. 5% of subjects reported the time spent was too much. In total, 97% of subjects and their parents would participate in similar studies in the future. Other responses in the end-of-study questionnaire regarding tolerability are displayed in S1 Table.

**Table 1. Baseline characteristics.**

| | Complete cohort |
|---|---|
| | (n = 175) |
| **Age (Mean (SD))** | 9.1 (4.3) |
| **Sex** | |
| Male | 45.7% |
| Female | 54.3% |
| **Ethnicity** | |
| Caucasian | 92% |
| Other/mixed | 8% |
| **Height (cm) (Mean (SD))** | 138.9 (26.1) |
| **Weight (kg) (Mean (SD))** | 37.2 (17.5) |
| **BMI SDS (Mean (SD))** | 0.1 (1.2) |
| **Daytime activity** | |
| Day care | 11% |
| Primary school | 50% |
| Secondary school | 37% |
| Vocational education | 1% |
| **Plays sports (%)** | 84.6% |
| **Population density** | |
| $< 1000 / km^2$ | 1% |
| $1000–1500 / km^2$ | 19% |
| $1500–2500 / km^2$ | 13% |
| $> 2500 / km^2$ | 67% |
| **PedsQL score (Mean (SD))** | 90.6 (7.3) |

Abbreviations: BMI SDS: body mass index standard deviation score

## Physical activity

**Physical activity per day.** 174 subjects provided at least one day of physical activity. Step count per day (Daily PA) was considered as the principal candidate endpoint. The relationship between age and step count was best described as a 3$^{rd}$ order natural spline (Fig 1A, ΔAIC 44, $p < 0.001$). On average, the daily step count of male subjects was 1082 steps higher compared to female subjects (95% CI 1609–556, $p < 0.001$, Fig 1B), although not for all ages (Fig 1G). Rain duration (ΔAIC 38, $p < 0.001$, Fig 1C) and ambient temperature (as 3$^{rd}$ order natural spline, ΔAIC 13, $p < 0.001$, Fig 1D) were also significantly associated with Daily PA. Furthermore, physical activity was lower on Sundays (Fig 1E), and for children with a lower population density of the city of residence compared to children in highly urbanized areas (difference 1199, 95% CI 578–1819, $p < 0.001$, Fig 1F). Finally, the average watch wear time between 6AM-10PM was also associated with step count per day (ΔAIC 230, $p < 0.001$). BMI, ethnicity, playing sports and QoL were not significantly associated with step count in this cohort. The model estimated intra-subject CV (corrected for factors rain, temperature, and day of the week) was 18%.

The number of steps taken during the most active hour per day (Hourly PA$^{max}$) was considered as a separate endpoint. The measurement was correlated to Daily PA (R = 0.8, $p < 0.001$), and identical variables explained variability. Model-estimated intra-subject CV was 23%, and the marginal R$^2$ of the multivariate model was 0.18 (Table 3). The 90% prediction interval stratified by sex is displayed in Fig 2A.

**Table 2. Compliance during the study period.**

| Measurement | Median compliance (IQR) |
|---|---:|
| Smartwatch | |
| Step count | 100% (100%–100%) |
| Heart rate | 100% (100%–100%) |
| Sleep | 95% (85%–100%) |
| Wear time per day | 23.6h (22.8h–23.9h) |
| Questionnaire | 90% (81%–100%) |
| Temperature | 83% (67%–100%) |
| Weight* | 100% (67%–100%) |
| Blood pressure* | 95% (85%–100%) |
| Spirometry* | 83% (67% -100%) |
| Overall compliance | 94% (87%–97% |

* Subjects ≥ 6 years old only

**Physical activity per week.** Daily step count was averaged per week (Daily $PA^{avg}$). Similar factors were associated with this candidate endpoint (Table 3). A multivariate model for weekly PA including age, sex, the interaction between age and sex, average temperature, average rain duration, average watch wear time between 6AM-10PM and population density of the subject's place of residence provided the best fit (marginal $R^2$ of 0.50). The model estimated intra-subject CV was 8%.

Four other endpoints based on physical activity within a week were considered and focused on peak- and trough values: the 10th and 90th percentile of daily PA within a week (Daily $PA^{10th}$ and Daily $PA^{90th}$) and the 50th and 90th percentile of step count per hour within a week (Hourly $PA^{50th}$ and Hourly $PA^{90th}$). Factors of influence on these endpoints are displayed in Table 3. Fig 2B–2F display the 90% prediction intervals of the proposed endpoints.

The estimated sample size necessary to detect a 15% increase in a parallel study with 21 days of measurements with correction for the factors of influence ranged from 33–110 per group for all candidate endpoints related to physical activity. All candidate endpoint characteristics are listed in Table 3. Final model coefficients for physical activity-related endpoints are displayed in S2 Table.

## Heart rate

HR data was obtained from 170 subjects. Daytime- and nighttime HR were associated with age (marginal $R^2$ = 0.62, p < 0.001 for daytime HR and marginal $R^2$ = 0.50, p < 0.001 for nighttime HR). The relationship was best described with a 3rd degree spline (Fig 3A). The population mean and 90% prediction interval were within the reference 10th-90th centile of resting HR obtained from the literature [16]. On average, female HR was 2.7 bpm higher compared to male HR (95% CI 1.0–4.4, p = 0.002). Intra-subject CV was 6% (daytime HR) and 8% (nighttime HR). Fig 3B shows the average HR during the day for age-groups 2–5, 6–12 and 13–16. There was a statistically significant correlation between HR and step count, although the effect varied by age. Average daytime HR increased by 1.1 bpm (95% CI 0.9–1.3) on average for every 1000 steps taken by 16-year-old children, while increasing 0.5 bpm (95% CI 0.4–0.6) per 1000 steps for 8-year-old children (Fig 3C). Model coefficients are listed in S3 Table.

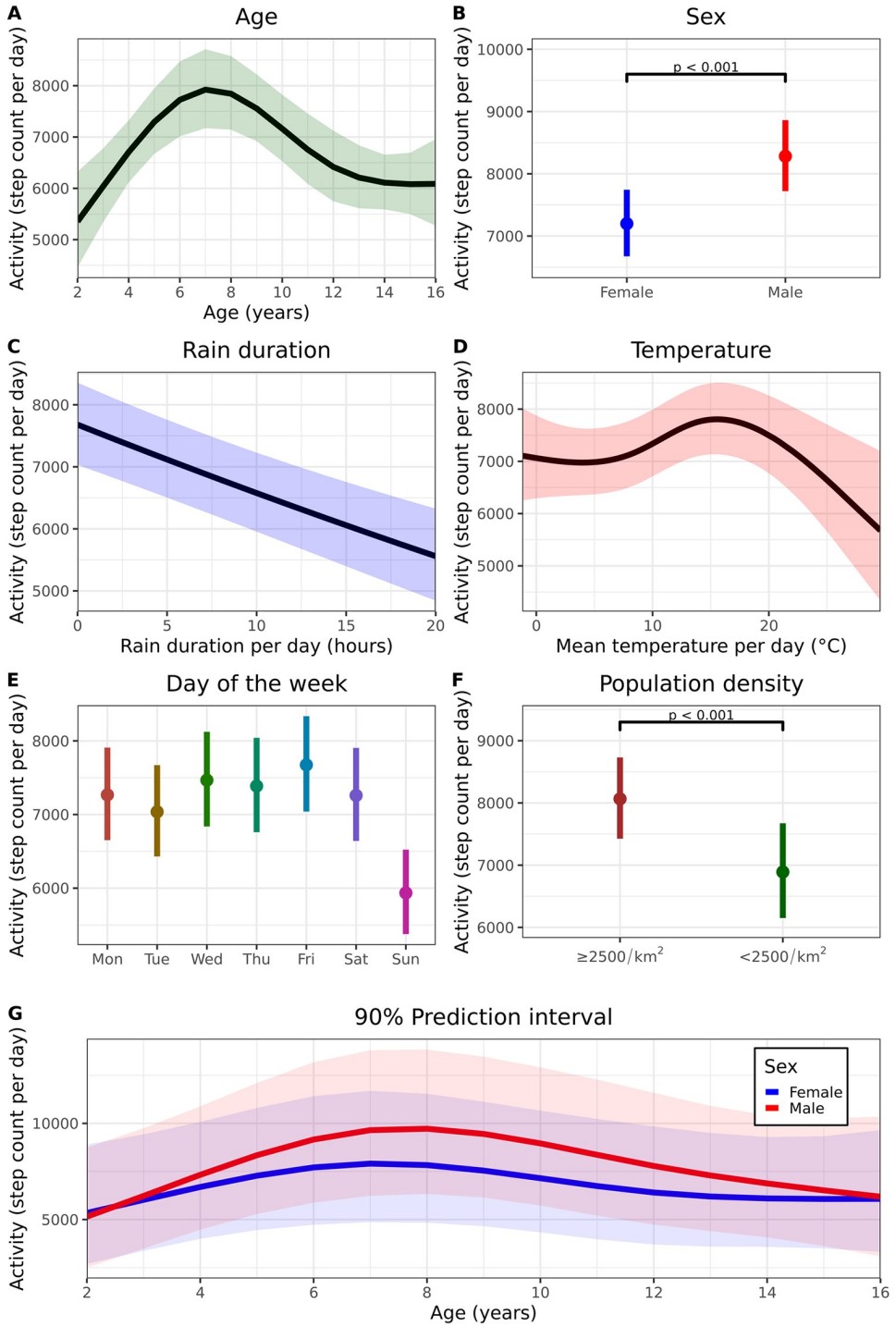

**Fig 1. Factors related to daily physical activity and reference values.** A. Relationship between mean daily physical activity (95% CI) and age. B. Relationship between daily PA and sex. The interaction between age and sex was excluded from the model for this graph only. C. Relationship between daily rain duration and physical activity. D: estimated mean (95% CI) daily PA with varying ambient temperature. E: estimated mean (95% CI) physical activity for each day of the week. F: estimated mean daily PA for children living in a highly urbanized area ($>$ 2000 people/km$^2$) or a less urbanized area ($<$ 2000 people/km$^2$). G. 90% prediction interval of daily PA stratified by age and sex. Wear time (100%), rain duration (2h) and temperature (11˚C) are held constant. Personalized predictions can be made using the model coefficients in S4 Table.

**Table 3. Physical activity-related candidate endpoints.**

| Sampling frequency | Candidate endpoint | Associated factors* | Intra-subject CV** | Marginal R² / Conditional R² | ICC | Prediction interval | Sample size needed to detect 15% increase*** |
|---|---|---|---|---|---|---|---|
| Per day | Daily PA | Age, sex, rain duration, temperature, day of the week, population density, watch wear time | 18% | 0.29 / 0.47 | 0.25 | Fig 1G | n = 37 |
| | Hourly PA^max | | 23% | 0.18 / 0.30 | 0.15 | Fig 2A | n = 35 |
| Per week | Daily PA^avg | Age, sex, mean rain duration, mean temperature, mean watch wear time, population density | 8% | 0.50 / 0.78 | 0.58 | Fig 2B | n = 35 |
| | Daily PA^90th | | 9% | 0.46 / 0.72 | 0.47 | Fig 2C | n = 33 |
| | Daily PA^10th | | 12% | 0.39 / 0.74 | 0.57 | Fig 2D | n = 67 |
| | Hourly PA^90th | | 9% | 0.46 / 0.76 | 0.55 | Fig 2E | n = 38 |
| | Hourly PA^50th | Age, sex, watch wear time | 5% | 0.24 / 0.71 | 0.62 | Fig 2F | n = 110 |

Abbreviations: CV: coefficient of variability, ICC: intraclass correlation coefficient, PA: physical activity.

* final model coefficients are displayed in S2 Table.

** Adjusted for associated factors

*** Approximate sample size needed per group to be able to detect a 15% increase with 80% power in a hypothetical parallel group study where subjects are monitored for 21 consecutive days and the outcome is adjusted for associated factors.

## Sleep parameters

Accelerometer-derived nocturnal sleep parameters were obtained from 172 subjects. Sleep duration decreased as a function of age (ΔAIC 65, p < 0.001, Fig 4A), and was similar across weekdays, except for weekends of older children (S3 Fig). Average percentage of light sleep was higher as subjects got older (increase of 0.29% per age year (95% CI 0.08–0.51, p = 0.006) (Fig 4B). On average, the proportion of light sleep of female subjects was 2.5% (95% CI 0.7–4.3, p = 0.008) lower than sleep depth of male subjects. Day of the week was not related to sleep

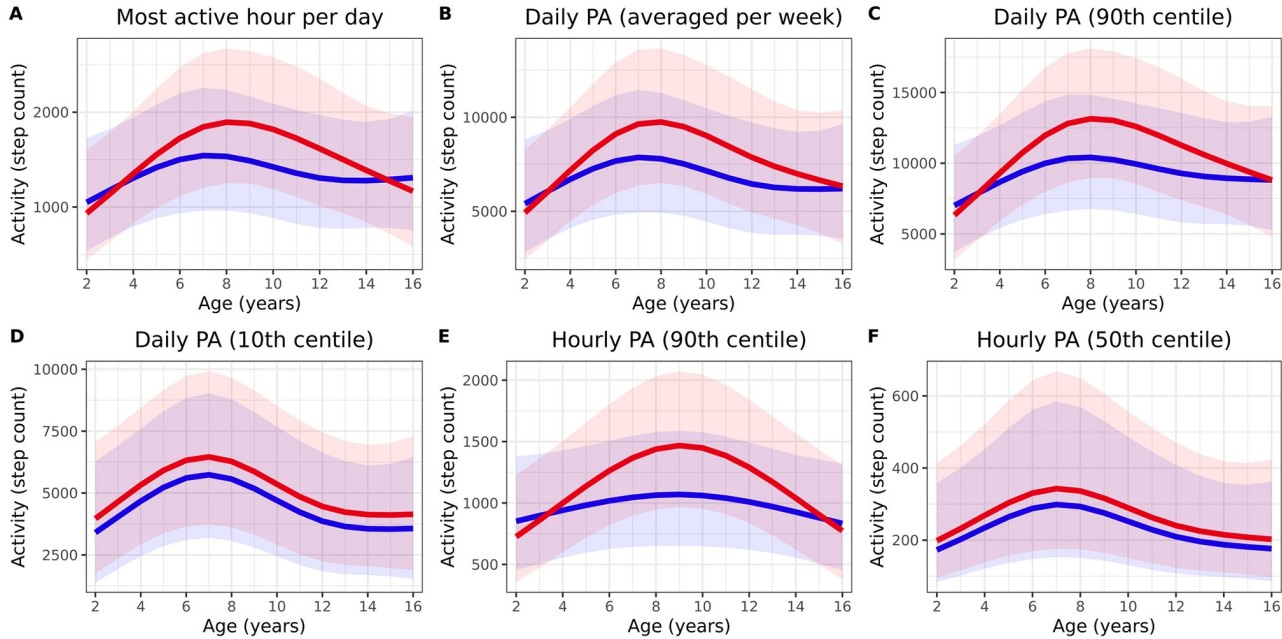

**Fig 2. Prediction interval of physical activity candidate endpoints.** Prediction interval of several physical-activity related candidate endpoints. Red and blue line represent the estimated mean for male subjects and female subjects, respectively. The shaded areas represent the 90% prediction intervals where watch wear time (100%), rain duration (2h) and temperature (11°C) are held constant.

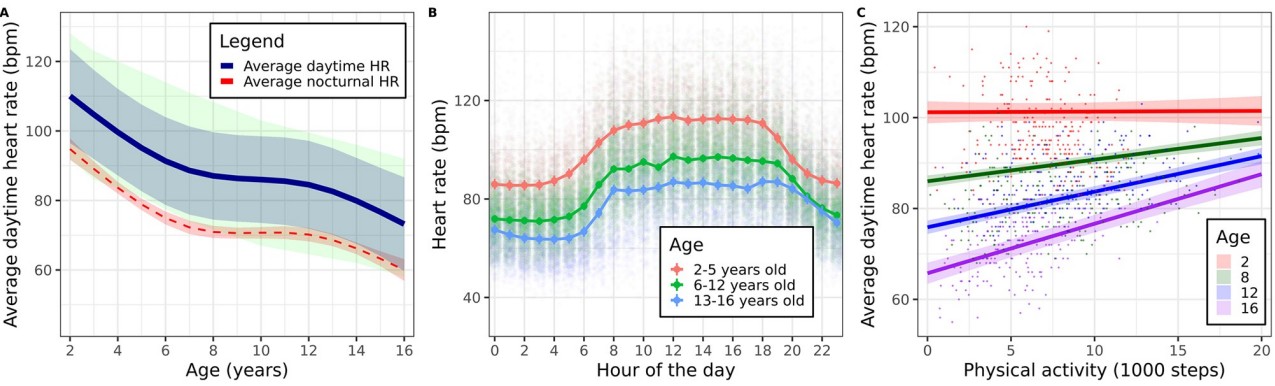

**Fig 3. Heart rate.** A: average (90% prediction interval) HR during daytime (blue). Average 95% CI nighttime HR (red). Reference values (10$^{th}$– 90$^{th}$ percentile of HR per age year) are displayed in green. B: Average HR (95% CI) per hour of the day. C: estimated relationship between HR and step count for ages 2, 8, 12 and 16. A limited amount of age years is shown for readability.

depth. On average, older subjects woke up less often compared to younger subjects, although the correlation was weak (p < 0.001, marginal R$^2$ 0.06, Fig 4C). There was no statistically significant difference in wakeup count between male and female subjects (difference -0.07, 95% CI -0.4–0.3, p = 0.70), and there was no correlation between ambient temperature and wakeups. Model coefficients are listed in S4 Table.

## Spirometry, blood pressure and temperature

747 spirometry sessions were performed by 126 subjects during the study. 322 sessions (43%) were considered of sufficient quality (ATS grade A-C) for further analysis. Of these, the mean percentage of predicted FEV1 was 94% (95% CI 91–96%) and the mean percentage of predicted FVC was 99% (95% CI 96–102%). Of 2219 BP measurements, mean intra-subject CV was 7% for systolic BP and 10% for diastolic BP. S4 Fig graphically displays BP measurements per age year. Temperature measurements are presented in S5 Fig.

## Discussion

There is a need for novel pharmacodynamic clinical endpoints in pediatrics [19]. Continuous monitoring of physical parameters with digital- or wearable devices has potential as digital

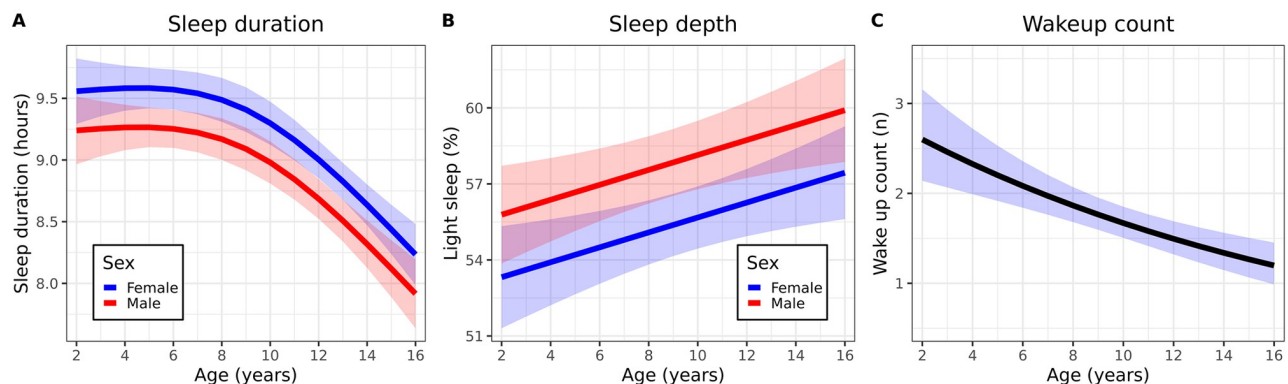

**Fig 4. Accelerometer-derived sleep parameters.** A: mean (95% CI) sleep duration stratified by sex. B: mean (95% CI) sleep depth stratified by sex. C: mean (95% CI) wakeup count.

endpoint in clinical research and patient care, due to the non-invasive and home-based nature of the measurements. However, extensive clinical validation in the target population is necessary before implementation. Because clear guidelines regarding validation of digital endpoints were lacking, our consortium recently published a pragmatic approach towards fit-for-purpose clinical validation [4]. This study focuses on several crucial steps of this validation process. In the first place, the tolerability of study devices for the target population is important to assess and, if necessary, improve. Secondly, the repeatability of candidate digital endpoints in free-living conditions should be investigated before integration in clinical trials. Furthermore, reference values of candidate digital endpoints in the target population should be obtained and, ideally, datasets should be shared to avoid unnecessary duplication.

Compliance is an important indicator of the tolerability of the clinical trial regimen, and even more so for clinical care, where the follow-up period may be indefinite. In this study, the median compliance was 94%. Based on the literature, a compliance of 70% was determined as a cut-off value to define noncompliant subjects [20, 21]. Only 6% of subjects exhibited a compliance lower than this, and the high tolerability was also reflected in the responses in the EOS questionnaire: 97% of subjects and their parents indicated they would participate in similar clinical studies in the future. Smartwatch-related measurements exhibited the highest compliance and may be most suitable to investigate further for their utility in clinical care. The high frequency of smartwatch-related measurements may provide a high-resolution overview of patients' disease state, while objectivity is another advantage compared to traditional questionnaire-based disease scores. Besides subjectivity, questionnaires also suffer from recall bias.

This study investigated the repeatability of candidate endpoints related to physical activity and assessed conditions that explain the observed variability in free-living conditions. Adjusting for such factors in clinical trials will improve statistical power and reduce required sample sizes. For all candidate endpoints, similar covariates and factors were found to explain variability and included age, sex, rain duration, temperature, population density and watch wear time. Some of the reported relationships (age, sex) with physical activity have been reported extensively the past as well [22], including the relationship between population density and step count [23]. This relationship may be explained by children cycling more in less densely populated areas, which is not registered as steps by wrist-worn smartwatches. Interestingly, overall step counts were lower than reported in the literature [22]. This may be partly explained by inter-device variability or by the trend towards a more sedentary lifestyle. BMI, playing sports and QoL (PedsQL questionnaire) were not related to physical activity. However, considering this was a cohort with healthy participants, the range of values of BMI and QoL was small and as such, an association may not have been expected [12].

Candidate endpoints based on physical activity can be divided in three groups: general- (e.g. step count per day), peak- (e.g. hourly $PA^{max}$) and trough physical activity (e.g. weekly $PA^{10th}$). We hypothesize that a decrease in peak or trough physical activity may be an early sign of worsening disease activity in chronic disease and therefore be promising for follow-up in clinical care. In the past, research focus has been mainly on moderate-to-vigorous physical activity (MVPA). MVPA cannot easily be derived from step count, although the two are highly correlated [24]. On the other hand, MVPA completely discounts light-intensity activity and is often planned (e.g. gym class, sports). Light-intensity is usually unplanned, more voluntary, and may contain important information relating to disease-activity.

In the end, the optimal endpoint related to activity will be determined by their ability to discriminate healthy from ill children, their association with symptom severity, and the measurement purpose. For example, a high sampling rate of a measurement with a slightly higher intra-subject variability, such as $PA^{max}$ will still lead to a precise estimation of the group mean in a clinical trial, while endpoints with lower intra-subject variability, such as weekly $PA^{avg}$,

will be more suitable for individual follow-up. Clinical validation in pediatric patient groups should be performed to identify the most suitable endpoints for the different aims. With the exception of endpoints based on trough physical activity, calculated sample sizes needed to detect a 15% increase in candidate endpoints appear feasible.

This study aimed to obtain reference values in healthy children for candidate digital endpoints related to physical activity in the form of 90% prediction intervals. While these graphically presented prediction intervals could be used as a screening tool for pediatric patients, individual predictions that take weather condition, wear time and city of residence into account may be more appropriate and can be calculated using the model coefficients in S2 Table.

HR has been proposed as a candidate biomarker in, among others, pediatric pulmonology, intensive care and psychiatry [25–27], and remote non-invasive HR monitoring could extend this measurement to the home-setting. In the past, Pelizzo et al. have shown that HR measured via PPG technology can be reliable in children [28]. Although performed in a surgery setting, direct comparison to ECG-derived HR demonstrated reliability. In this study, the average heart rate in free-living conditions was compared to known reference values in pediatrics and showed good concordance. Furthermore, a subtle but statistically significant difference in mean HR was found between boys and girls, a finding that has been reported before [29, 30]. The lower nocturnal HR compared to daytime HR [31], as well as the correlation that was found between average daytime HR with step count was expected but provides another indication of the validity of PPG measurements in a pediatric population in free-living conditions. The absence of this positive correlation in younger children has been reported before by Herzig et al. [32].

Accelerometer-derived sleep duration and depth have been shown to be less accurate when compared to polysomnography, which is the gold standard. However, average total sleep duration in this study appeared to correlate well with published nighttime sleep duration norms [17], and new endpoints for use in free-living conditions should be compared to current standards in the home-setting, such as, comparably reliable, sleep-diaries [33]. Nevertheless, careful interpretation of sleep duration, especially in preschool subjects is necessary. At least three hours of inactivity are needed to register sleep, and daytime naps often do not meet this criterion [34]. Comparison of sleep parameters with pediatric patients with known difficulty sleeping or nocturnal unrest could be helpful to determine the usability of sleep parameters obtained from smartwatches. For example, patients with ARID1B-related intellectual disability showed a higher frequency of wakeups in a recent study [35], and similar differences could be expected in attention deficit hyperactivity disorder, or pediatric asthma.

Temperature, BP, and spirometry measurements are well established measurements with known normative values and are routinely employed in a clinical trial setting. In this study, the measurements were generally within normal range. However, although the used BP monitor has been validated in adults, formal validation has not been performed in children. As a result, clinical interpretation of the measurements should be done with extreme care. However, tolerability and usability of Bluetooth connected devices in a home-setting has not been investigated before and was sufficient in this study, although compliance was slightly lower compared to measurements with a smartwatch.

This study has several limitations. Although the study period of three weeks was long compared to other studies, it was relatively short for the purpose of simulating a possible clinical trial regimen. Compliance to study tasks may decrease with longer follow-up periods. While many variables were collected and related to physical activity, it is possible that a portion of unexplained variability could be accounted for by factors such as socio-economic-status of the parents, which was not registered in this study. Future studies may investigate the influence of variables such as these to increase our understanding of the drivers of physical activity and

allow for better isolation of disease- or treatment effects. Conversely, factors that were not registered in this study could change the estimated effects between age and physical activity or heart rate. However, the observed effects of age are plausible and have been reported in the literature in the past. Children were instructed to wear the watch at all time, including during sports. However, it is possible that subjects took off the watch, for example, during contact sports. This may negatively impact the total step count per day. Although adjustment for watch wear time was performed during the statistical analysis, no information regarding the exact reason of missing data was available, and could be either due to not wearing the watch, connectivity errors between the smartwatch and smartphone, or inappropriate handling of the device during the collection or transfer of data. However, this lack of information regarding the cause of missing data is inherent to the field of remote monitoring and finding statistical methods to adjust for this discrepancy is important. During this study, the watch wear time was derived by appraising both the heart rate and step count data during each individual hour. If either was registered, it was determined that the watch was worn during that hour. However, a mismatch was observed between step count and HR data. Some days included more hours with viable HR data than step count data and vice versa. In the case of missing HR data, this discrepancy could be due to an inadequate position of the smartwatch on the wrist of the child, possibly during movement. While this particular mismatch and the general presence of missing data is suboptimal compared to conventional supervised measurements in the clinic, the fact that the size of the current dataset is much larger compared to conventional trials may outweigh the disadvantages of randomly missing data points. Still, out of the possible 88,200 hours and 3675 days of step count and HR data, between 91% (HR per hour) and 96% (day with wear time > 50%) of observations were included in the final analysis set, which indicates that the impact of missing data on the overall conclusions is likely negligible [36]. The compliance cut-off of 50% wear time per day is lower than employed in other studies in adults and could be revisited in future analyses. It was estimated a priori that children, especially the youngest, would find it difficult to wear the watch 24 hours a day and using this lower cut-off in combination with the statistical adjustment for wear time was expected to lead to valid conclusions. Increasing the threshold to a higher range, for example 70% would lead to exclusion of only a small number (2%) of additional observations. Future clinical validation in patient groups is necessary to confirm whether the employed analysis methods can detect the effects of treatment and changes in disease-activity despite the potential impact of missing data.

To our knowledge, this study is the first to investigate multiple smartwatch- and digital health-related measurements in a pediatric cohort of this size and age-range. The field of digital endpoints is relatively undeveloped and data sharing may accelerate development and avoid unnecessary duplication. S1 Dataset contains the complete dataset that was used during analysis. A legend of included variables is listed in S5 Table. Future research may focus on tolerability and compliance during a longer follow-up period and further clinical validation of the proposed endpoints in pediatric patient groups.

## Conclusion

The investigated home-monitoring platform with a range of wearables and other home-monitoring devices has a good tolerability and led to high compliance. We propose several candidate endpoints related to physical activity that could be used in pediatric trials. Observed variability in endpoints was largely explained by a combination of age, sex, and weather circumstances. In the future, the reference values provided for the candidate endpoints could be used in clinical care and for clinical trial design. Heart rate and sleep data provided by the smartwatch are comparable to pediatric reference values and appear a valid option for

pediatric clinical trials in a home-setting. Further clinical validation in pediatric patient groups is currently ongoing and necessary to further define the value of the proposed digital endpoints.

## Supporting information

**S1 Fig. Median (IQR) compliance to study tasks by age.**
(PDF)

**S2 Fig. Decrease in compliance over time.**
(PDF)

**S3 Fig. Sleep duration by day of the week.**
(PDF)

**S4 Fig. Blood pressure measurements per age year.**
(PDF)

**S5 Fig. Temperature measurements per age year.**
(PDF)

**S1 Table. Tolerability questionnaire.**
(PDF)

**S2 Table. Model coefficients of physical activity-related candidate endpoints.**
(PDF)

**S3 Table. Model coefficients of heart rate parameters.**
(PDF)

**S4 Table. Model coefficients of accelerometer-derived sleep parameters.**
(PDF)

**S5 Table. Dataset legend.**
(PDF)

**S1 Data. Complete dataset generated during this study.**
(CSV)

**S1 File. R code used during analysis.**
(R)

**S2 File. End-of-study questionnaire.**
(PDF)

## Acknowledgments

The authors wish to thank the clinical trial assistants and data management officers for their contribution to the study, Pip Lambrechtse for participation during protocol design, Esmée Essers for logistical support, as well as all study participants and their parents for their enthusiasm during the conduct of this study.

## Author Contributions

**Conceptualization:** M. D. Kruizinga, M. L. de Kam, F. E. Stuurman, A. F. Cohen, G. J. A. Driessen.

**Data curation:** M. D. Kruizinga, A. Zhuparris.

**Formal analysis:** M. D. Kruizinga, N. van der Heide, A. Zhuparris, Y. Yavuz, M. L. de Kam.

**Investigation:** M. D. Kruizinga, N. van der Heide, A. Moll, F. E. Stuurman, G. J. A. Driessen.

**Methodology:** M. D. Kruizinga, A. Moll, F. E. Stuurman, A. F. Cohen, G. J. A. Driessen.

**Project administration:** M. D. Kruizinga, N. van der Heide.

**Supervision:** Y. Yavuz, M. L. de Kam, F. E. Stuurman, A. F. Cohen, G. J. A. Driessen.

**Visualization:** M. D. Kruizinga, A. Zhuparris.

**Writing – original draft:** M. D. Kruizinga.

**Writing – review & editing:** N. van der Heide, A. Moll, A. Zhuparris, Y. Yavuz, M. L. de Kam, F. E. Stuurman, A. F. Cohen, G. J. A. Driessen.

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
