## [Decision Letter · Decision Letter 0]

16 Nov 2020

PONE-D-20-30878

Towards remote monitoring in pediatric care and clinical trials – Tolerability, repeatability and reference values of candidate digital endpoints derived from physical activity, heart rate and sleep in healthy children.

PLOS ONE

Dear Dr. Kruizinga,

Thank you for submitting your manuscript to PLOS ONE. After careful consideration, we feel that it has merit but does not fully meet PLOS ONE’s publication criteria as it currently stands. Therefore, we invite you to submit a revised version of the manuscript that addresses the points raised during the review process.

Please address the reviewers' comments with special emphasis on the ones listed below:

1. Full description of the statistical methodology used and the results reported.

2. Discussion on the wear time and its influence on the analyses undertaken.

3. Validity of the data obtained from the Steel HR smartwatch.

We look forward to receiving your revised manuscript.

Kind regards,

Jaroslaw Harezlak, PhD

Academic Editor

PLOS ONE

Journal Requirements:

2.In your Methods section, please provide additional information about the participant recruitment method and the demographic details of your participants. Please ensure you have provided sufficient details to replicate the analyses such as: a) a description of any inclusion/exclusion criteria that were applied to participant recruitment, b) descriptions of where participants were recruited.

Reviewers' comments:

Reviewer's Responses to Questions

**Comments to the Author**

1. Is the manuscript technically sound, and do the data support the conclusions?

Reviewer #1: Yes

Reviewer #2: Partly

2. Has the statistical analysis been performed appropriately and rigorously? 

Reviewer #1: Yes

Reviewer #2: Yes

3. Have the authors made all data underlying the findings in their manuscript fully available?

Reviewer #1: Yes

Reviewer #2: Yes

4. Is the manuscript presented in an intelligible fashion and written in standard English?

Reviewer #1: Yes

Reviewer #2: Yes

5. Review Comments to the Author

Reviewer #1: This study examines the compliance and tolerability of using wearable and portable devices for home monitoring in healthy children between 2 and 16 years old. In addition, the study aims to determine reference values of candidate digital endpoints derived from physical activity, hear rate and sleep data in this population. This work presents an effort towards remote monitoring in pediatric population and holds the potential to provide useful guide for future clinical trials. My major comments are provided below.

1. In the Analysis Section (page 5, line 115), the calculation of the tolerability was discussed. However, little description was provided regarding how the repeatability was quantified in the paper.

2. In the Results Section, please provide test statistic, p value, and effect size for the findings in the study. For example, when reporting rain duration (Figure 1C) and ambient temperature (Figure 1D) were significantly associated with Daily PA (page 9, line 206), no test statistics were reported to support the conclusion. Similarly, when reporting sleep parameters (lines 253-262), test statistic, p value, effect size and confidence interval should be reported in a consistent way throughout the paper. When reporting a statistically significant correlation between HR and step count (line 248), test statistic and p value should be provided in the text to support the conclusion.

3. Table 3 includes 7 candidate endpoints, each of which was examined with respect to multiple factors (e.g., age, sex, wear time). The large number of tests conducted could lead to inflated false positives. Results with multiple comparison correction should be reported.

4. The study recruited 10-15 children of each age between 2 and 16 years old (line 91). When examining the relationship between these endpoints and age (e.g., Figure 1A, Figure 3A), how could we be sure that the observed change was due to age rather than other potential differences between these different groups of children?

5. As the study collected data over 21 days, it will be helpful to know the compliance over time. Was there a decrease in compliance over time? This is important as clinical trials are likely to require a longer time of using wearable and portable devices.

6. Did participants receive any compensation for completing these measures (e.g., temperature, weight, and spirometry measurements)? If so, details should be reported. Strategies on using incentives to improve compliance would be helpful for future clinical trials.

Reviewer #2: The manuscript is technically sound, conditional on details relating to the underlying wearable device data. My specific comments in this regard have been uploaded as an attached pdf document.

6. PLOS authors have the option to publish the peer review history of their article (what does this mean?). If published, this will include your full peer review and any attached files.

Reviewer #1: No

Reviewer #2: No

---

## [Author Response · Author response to Decision Letter 0]

20 Nov 2020

Dear editor and reviewers,

We thank you for reading our manuscript and your thoughtful feedback. We have considered your comments carefully and implemented your requests wherever possible. Please refer to our detailed response to your questions below at the end of this PDF file. 

Kind regards,

Matthijs Kruizinga

Gertjan Driessen

---

## [Decision Letter · Decision Letter 1]

15 Dec 2020

PONE-D-20-30878R1

Towards remote monitoring in pediatric care and clinical trials – Tolerability, repeatability and reference values of candidate digital endpoints derived from physical activity, heart rate and sleep in healthy children.

PLOS ONE

Dear Dr. Kruizinga,

Thank you for submitting your manuscript to PLOS ONE. After careful consideration, we feel that it has merit but does not fully meet PLOS ONE’s publication criteria as it currently stands. Therefore, we invite you to submit a revised version of the manuscript that addresses the points raised during the review process.

Please address one remaining major issue on the wear-time estimation and additional minor comments.

We look forward to receiving your revised manuscript.

Kind regards,

Jaroslaw Harezlak, PhD

Academic Editor

PLOS ONE

Reviewers' comments:

Reviewer's Responses to Questions

**Comments to the Author**

1. If the authors have adequately addressed your comments raised in a previous round of review and you feel that this manuscript is now acceptable for publication, you may indicate that here to bypass the “Comments to the Author” section, enter your conflict of interest statement in the “Confidential to Editor” section, and submit your "Accept" recommendation.

Reviewer #1: All comments have been addressed

Reviewer #2: (No Response)

2. Is the manuscript technically sound, and do the data support the conclusions?

Reviewer #1: Yes

Reviewer #2: Partly

3. Has the statistical analysis been performed appropriately and rigorously? 

Reviewer #1: Yes

Reviewer #2: Yes

4. Have the authors made all data underlying the findings in their manuscript fully available?

Reviewer #1: Yes

Reviewer #2: Yes

5. Is the manuscript presented in an intelligible fashion and written in standard English?

Reviewer #1: Yes

Reviewer #2: Yes

6. Review Comments to the Author

Reviewer #1: (No Response)

Reviewer #2: I have uploaded my Major and Minor comments as an attached pdf document for this resubmission.

7. PLOS authors have the option to publish the peer review history of their article (what does this mean?). If published, this will include your full peer review and any attached files.

Reviewer #1: No

Reviewer #2: No

---

## [Author Response · Author response to Decision Letter 1]

17 Dec 2020

Dear editor and reviewer,

Please find the response to your comments at the end of the submission file.

Kind regards, Matthijs Kruizinga

---

## [Editor Report · Decision Letter 2]

18 Dec 2020

Towards remote monitoring in pediatric care and clinical trials – Tolerability, repeatability and reference values of candidate digital endpoints derived from physical activity, heart rate and sleep in healthy children.

PONE-D-20-30878R2

Dear Dr. Kruizinga,

We’re pleased to inform you that your manuscript has been judged scientifically suitable for publication and will be formally accepted for publication once it meets all outstanding technical requirements.

Kind regards,

Jaroslaw Harezlak, PhD

Academic Editor

PLOS ONE

Additional Editor Comments (optional):

A minor mistake: "Aikaike" should be replaced by "Akaike" in the AIC definition in the Abbreviations table.
---

## [Editor Report · Acceptance letter]

26 Dec 2020

PONE-D-20-30878R2 

Towards remote monitoring in pediatric care and clinical trials – Tolerability, repeatability and reference values of candidate digital endpoints derived from physical activity, heart rate and sleep in healthy children. 

Dear Dr. Kruizinga:

I'm pleased to inform you that your manuscript has been deemed suitable for publication in PLOS ONE. Congratulations! Your manuscript is now with our production department. 

Kind regards, 

on behalf of

Dr. Jaroslaw Harezlak 

Academic Editor

PLOS ONE